# Inflammatory Response in the Pathogenesis and Treatment of Hepatocellular Carcinoma: A Double-Edged Weapon

**DOI:** 10.3390/ijms25137191

**Published:** 2024-06-29

**Authors:** Linda Galasso, Lucia Cerrito, Valeria Maccauro, Fabrizio Termite, Irene Mignini, Giorgio Esposto, Raffaele Borriello, Maria Elena Ainora, Antonio Gasbarrini, Maria Assunta Zocco

**Affiliations:** 1Department of Internal Medicine and Gastroenterology, Fondazione Policlinico Universitario Agostino, Gemelli IRCCS, Catholic University of Rome, 00168 Rome, Italy; linda.galasso@guest.policlinicogemelli.it (L.G.); lucia.cerrito@policlinicogemelli.it (L.C.); valeria.maccauro@guest.policlinicogemelli.it (V.M.); fabrizio.termite@libero.it (F.T.); irene.mignini@guest.policlinicogemelli.it (I.M.); giorgio.esposto@guest.policlinicogemelli.it (G.E.); raffaele.borriello01@icatt.it (R.B.); mariaelena.ainora@policlinicogemelli.it (M.E.A.); antonio.gasbarrini@policlinicogemelli.it (A.G.); 2CEMAD Digestive Disease Center, Fondazione Policlinico Universitario Agostino Gemelli IRCCS, Catholic University of Rome, 00168 Rome, Italy

**Keywords:** hepatocellular carcinoma, inflammation, viral, fatty liver

## Abstract

Hepatocellular carcinoma (HCC) is the most frequent among primary liver tumors (90%) and one of the main causes of cancer-related death. It develops usually in a chronically inflamed environment, ranging from compensatory parenchymal regeneration to fibrosis and cirrhosis: carcinogenesis can potentially happen in each of these stages. Inflammation determined by chronic viral infection (hepatitis B, hepatitis C, and hepatitis delta viruses) represents an important risk factor for HCC etiology through both viral direct damage and immune-related mechanisms. The deregulation of the physiological liver immunological network determined by viral infection can lead to carcinogenesis. The recent introduction of immunotherapy as the gold-standard first-line treatment for HCC highlights the role of the immune system and inflammation as a double-edged weapon in both HCC carcinogenesis and treatment. In this review we highlight how the inflammation is the key for the hepatocarcinogenesis in viral, alcohol and metabolic liver diseases.

## 1. Introduction

Hepatocellular carcinoma (HCC) is the primary form of liver cancer, accounting for 80–90% of cases globally. It ranks seventh in terms of incidence worldwide and sixth as the most frequent cancer, while being the third-leading cause of cancer-related deaths [1,2]. Predictions indicate a staggering 55% increase in new liver cancer cases by 2040, potentially affecting 1.4 million individuals [3]. Furthermore, liver cancer is projected to cause over a million deaths annually by 2030 [4]. These statistics underscore the continued significance of HCC as a major global health concern. The majority of HCC cases arise in individuals with underlying liver disease, primarily cirrhosis, and associated risk factors such as chronic viral hepatitis infections (hepatitis B virus (HBV), hepatitis D virus (HDV) or hepatitis C virus (HCV)), alcohol-related liver disease (ALD), and metabolic dysfunction-associated steatotic liver disease (MASLD) [5]. Additional, albeit less common, risk factors include primary biliary cholangitis, haemochromatosis, and α1-antitrypsin deficiency [6].

Therefore, it can be inferred that chronic inflammation, regardless of its cause, represents the most significant risk factor for HCC [7] (Figure 1).

While chronic HBV and HCV infections, along with alcohol-related liver disease, historically represented the majority of HCC cases, advancements in viral hepatitis prevention and treatment have significantly alleviated their impact. Notably, HBV vaccination and effective therapies for chronic HBV and HCV infections have played a crucial role in this regard. Despite HBV infection remaining the primary global risk factor for HCC (33%), particularly prevalent in East and Southeast Asia, where it contributes to over half of all HCC cases, there are shifting epidemiological trends [8,9]. Alcohol abuse (30%) and HCV (21%) also continue to be significant contributors to HCC incidence [8]. The dynamics of HCC are shifting amidst rapid economic expansion, heightened alcohol intake, sedentary behaviors, and the rising prevalence of obesity and diabetes, both worldwide and particularly in Asia [10]. MASLD, closely associated with obesity and insulin resistance, is emerging as a significant catalyst in HCC progression. MASLD instigates chronic inflammation, disturbs lipid metabolism, and fosters a carcinogenic milieu conducive to HCC development. Its prevalence is anticipated to affect a quarter of the general populace and is forecast to surge by 56% between 2016 and 2030 [11,12].

Hepatocellular carcinoma (HCC) is a prime example of inflammation-associated cancer, with more than 90% of cases developing amongst chronic hepatic inflammation [13].

In HCC, a vicious cycle ensues where immune system action is compounded by a perpetuating inflammatory process driven by exogenous factors such as pathogen-associated molecular patterns (PAMPs) from pathogens or damage-associated molecular patterns (DAMPs) from necrotic cells [13]. Hepatic cells express a plethora of pattern recognition receptors (PRRs) that, upon interaction with DAMPs, play pivotal roles in liver function and disease progression [14]. Among these PRRs, Toll-like receptors (TLRs), for instance, are responsible for activating proinflammatory signaling pathways that connect chronic inflammation to HCC. These pathways include NF-κB, c-Jun N-terminal kinase/AP1, ERK, and p38. HMGB1, a prominent DAMP, engages with PRRs like TLR2, TLR4, TLR9, and RAGE, leading to the activation of multiprotein complexes such as inflammasomes (e.g., NLRP3) [14]. These complexes subsequently initiate the synthesis of inflammatory cytokines such as IL-1β and IL-18. This sustained inflammatory cascade facilitated by HMGB1 contributes to the invasion and metastasis of HCC cells [15]. Furthermore, anticancer therapies induce immunogenic cell death, releasing additional DAMPs that interact with TLRs, thereby perpetuating a cycle of inflammation and immune response. This cycle can either suppress or promote tumor growth depending on the specific context [16,17].

In the premalignant phase of hepatocarcinogenesis, continuous activation of inflammatory signaling pathways results in the generation of reactive oxygen species (ROS) and reactive nitrogen species (RNS) [13]. These factors, along with chronic compensatory hepatocyte regeneration, proliferation-induced mutagenesis, and compromised immune surveillance, play a critical role in tumor initiation and progression [18]. Additionally, oxidative stress contributes to increased cell migration, invasion, and metastasis in HCC. Under normal circumstances, oxidative stress is counteracted by antioxidant molecules such as superoxide dismutase (SOD), catalase (CAT), glutathione peroxidase (GPX), and peroxiredoxins, along with exogenous antioxidants like vitamin C and tocopherols [19]. However, in chronic liver diseases, the antioxidant capacity becomes overwhelmed.

Oxidative stress also induces the production of proinflammatory cytokines in Kupffer cells, which activates nuclear factor kappa-light-chain enhancer of activated B cells (NF-κB) pathways in hepatocytes, resulting in the overexpression of interleukin 6 (IL-6) and signal transducer and activator of transcription 3 (STAT3). Furthermore, ROS stimulate the release of profibrogenic cytokines, leading to the activation of hepatic stellate cells (HSCs) and the secretion of collagen into the extracellular matrix, thereby promoting fibrosis [20].

The mechanisms by which HBV, HCV, and HDV induce oxidative stress can be divided into three primary categories: disruption of mitochondrial function through calcium uptake, induction of endoplasmic reticulum (ER) stress and the unfolded protein response (UPR), and a virus-induced imbalance between ROS generation and scavenging. Here, we detail their roles in promoting inflammation and carcinogenesis [21]. We will also describe how metabolic dysregulations can contribute to hepatocarcinogenesis through an inflammation-induced mechanism.

## 2. Viral Etiological Factors Associated with Inflammation and Hepatocellular Carcinoma

### 2.1. Hepatitis B Infection

The hepatitis B virus (HBV), belonging to the *Hepadnaviridae family*, is hepatotropic and possesses a double-stranded DNA structure. Its entry into hepatocytes occurs via the sodium taurocholate co-transporting polypeptide (NTCP) receptor, a transporter specific to hepatocytes and bile acids [22]. Within the nucleus, HBV facilitates viral replication and transcription, alongside the dysregulation of cancer-related gene expression, including cyclin A2 (*CCNA2*), telomerase reverse transcriptase (*TERT*), myeloid/lymphoid or mixed-lineage leukemia 4 (*MLL4*), tumor protein P53 (*TP53*), catenin β1 (*CTNNB1*), lysine methyltransferase 2B (*KMT2B*), and cyclin E1 (*CCNE1*), through the integration of covalently closed circular DNA (cccDNA) into the hepatocyte genome [22] (Table 1).

This viral integration directly elevates the risk of liver cancer, even in non-cirrhotic patients, through mechanisms such as cis-mediated insertional mutagenesis, chromosomal instability, and the expression of aberrant viral proteins [23].

Globally, HBV accounts for approximately 33% of liver cancer cases, with regional variations in attributable risk. In regions like Africa and East Asia, HBV contributes to around 60% of liver cancer cases, whereas in the Western world, this figure hovers around 20%. Patients with untreated chronic hepatitis B face a 5-year expected cumulative incidence of cirrhosis ranging from 8% to 20%. Additionally, individuals with HBV-related cirrhosis have an annual risk of HCC ranging from 2% to 5% [24].

Chronic HBV infections can induce hepatocellular injury and turnover within an inflamed environment. Acting as non-cytopathic agents, they trigger persistent cellular damage by activating both innate and adaptive immune responses. Despite the initial effective clearance of infected hepatocytes by cytotoxic cells and antiviral cytokines like interferon γ (IFN-γ) and tumor necrosis factor α (TNFα), ongoing exposure to viral antigens and inflammation leads to the exhaustion or tolerance of antiviral T cells in chronic infections [25]. This exhaustion is marked by the expression of immune inhibitory receptors (e.g., programmed death 1 (PD-1), cytotoxic T-lymphocyte antigen 4 (CTLA-4), T-cell immunoglobulin, and mucin domain-containing protein 3 (TIM-3), the presence of regulatory T cells, cytokine dysregulation, and impairment of B cells. Within this context, components of the innate immune system, such as dendritic cells, Kupffer cells, and natural killer cells (NK cells), may perpetuate inflammation and hepatocellular injury in a cyclic manner [26]. Thus, it is not solely the virus, but rather the host’s immune system that bears responsibility for hepatocellular damage, despite the inability to deplete cccDNA. Therefore, continuous lifelong treatment is necessitated [27].

In HBV-related HCC, as evidenced by Lim C.J. et al. [28], an immunosuppressive environment within the tumor is established, aiming to mitigate host damage during prolonged viral infection. Inflammatory cells present in the infected liver, including NK cells, secrete a range of cytokines (e.g., interleukin 10 (IL-10), transforming growth factor β (TGFβ), interleukin 4 (IL-4), and interleukin 13 (IL-13)) and chemokines that facilitate fibrogenesis and tumorigenesis [29]. HBV exacerbates this by impairing NK cell cytotoxicity and survival, thereby perpetuating chronic hepatocellular injury and promoting epithelial-to-mesenchymal transition [30,31]. Conversely, several factors contribute to NK cell dysfunction, such as severe hypoxia, the accumulation of tumor cell metabolites, alterations in inhibitory receptors or NK inhibitors, and the presence of molecules like TGFβ, prostaglandin E2 (PGE2), or indoleamine 2,3-dioxygenase 1 (IDO1). Furthermore, immunosuppressive cytokines, including IL-4, IL-10, and IL-13, in the microenvironment exacerbate NK cell dysfunction. These collective factors contribute to the dysfunction of NK cells in HBV–HCC [32].

Kupffer cells and inflammatory monocytes also play a role in promoting the transition from chronic hepatitis to liver cancer by releasing proinflammatory cytokines such as TNFα, IL-6, and monocyte chemoattractant protein 1 (MCP-1). Additionally, they may cooperate with myeloid-derived suppressor cells (MDSCs) to induce CD8+ T-cell exhaustion [33].

HBV-specific CD8+ T cells exhibit crucial antiviral activity during acute infection, generating proinflammatory cytokines (e.g., IFN-γ, interleukin 2 (IL-2), and TNFα) and cytotoxic molecules (granzyme and perforin) to eliminate infected cells [34]. However, in chronic HBV infection, these cells persistently circulate, including hepatitis B surface antigen (HBsAg)-specific, hepatitis B core antigen (HBcAg)-specific, and HBV polymerase-specific CD8+ T cells. This persistence contributes to sustained hepatic inflammation, accelerating hepatocyte turnover, genetic mutations, and fibrosis, thereby promoting HCC development [35].

During the immune-tolerant phase, CD8+ T cells become dysfunctional, expressing markers of exhaustion (e.g., programmed cell death protein 1 (PD-1), cytotoxic T lymphocyte-associated protein 4 (CTLA-4), lymphocyte-activating 3 (LAG3), T-cell immunoglobulin and mucin domain-containing protein 3 (TIM-3), and T-cell immunoreceptor with Ig and ITIM domains (TIGIT)) under pathological stimuli such as high viral load, increased Treg cells, and immunosuppressive cytokines like IL-10 and TGFβ [36,37]. This defective anti-tumoral surveillance, combined with chronic inflammatory stimuli, contributes to HCC development and immune evasion in later stages of HBV infection. Moreover, exhausted CD8+ T cells contribute to maintaining chronic hepatic inflammation by producing inflammatory cytokines (IFN-α, TNFα, interleukin 17A (IL-17A), and interleukin 22 (IL-22)) in response to continuous stimulation from circulating HBV antigens [38,39].

Conversely, in chronic HBV infection, there is a transition from T-helper (Th1) CD4+ T cells to Th17 CD4+ T cells, along with a reduction in effector cytotoxic CD4+ T cells, in response to elevated viral load [40,41]. These changes contribute to ongoing hepatocellular injury and diminished anti-tumoral surveillance. Additionally, Treg cells expand during chronic HBV infection, stimulated by cytokines (e.g., TGFβ, IL-10, and C-C motif chemokine ligand 22 (CCL22)) secreted by other innate cells, to prevent excessive antiviral immune responses and establish an immunosuppressive microenvironment for HCC initiation and progression [42].

The buildup of HBV viral proteins, even in the absence of active viral replication, emerges as a significant factor in viral hepatocarcinogenesis. At the heart of HBV’s replication and hepatocarcinogenesis lies the HBV-encoded oncogene X protein (HBx), a key transcriptional product [43,44]. HBx exerts its influence by disrupting the regulation of hepatocellular cycle through diverse mechanisms. It impacts cell proliferation, death, transcription, and DNA repair pathways, thereby fostering genetic instability and neoplastic transformation. Specifically, HBx protein interacts with cyclic adenosine monophosphate (cAMP) response element-binding (CREB) protein/P300 (CBP/P300), directly modulating CREB-dependent transcription. Moreover, it engages cellular signaling pathways such as Ras/Raf, mitogen-activated protein kinase (MAPK), and Janus kinase (JAK)–signal transducer of activators of transcription (STAT), influencing transcriptional processes [45,46,47].

Additionally, HBx protein affects proteasomes, mitochondrial proteins, p53, and DNA damage-binding protein (DDB1), consequently eliciting apoptotic effects [48]. Several studies have demonstrated the involvement of HBx protein in the progression of hepatocellular carcinoma and the generation of metastasis, in both in vivo and in vitro settings [49,50].

HBc emerges as a pivotal contributor to the carcinogenic process. It promotes hepatocarcinogenesis by enhancing apoptosis resistance and stimulating hepatoma cell proliferation via activation of the Src/PI3K/Akt pathway [51]. Additionally, HBc collaborates synergistically with HBx, exacerbating liver cancer progression by suppressing the promoter activity of p53 [52].

**Table 1 ijms-25-07191-t001:** Molecular pathways associated with HBV infection and its progression to liver cancer.

Study	Molecular Pathway	Description
Zhao et al. [22]	NTCP receptor pathway	HBV entry into hepatocytes occurs via the NTCP receptor, which is specific to hepatocytes and bile acids.
Zhao et al. [22]	Integration of cccDNA	HBV integrates covalently closed circular DNA (cccDNA) into the hepatocyte genome, affecting cancer-related genes like CCNA2, TERT, MLL4, TP53, CTNNB1, KMT2B, and CCNE1.
Tu et al. [23]	Cis-mediated insertional mutagenesis	HBV integration into the genome causes insertional mutagenesis, chromosomal instability, and expression of aberrant viral proteins, increasing the risk of liver cancer.
Wangensteen et al. [25]	Immune response pathways	Chronic HBV infections trigger persistent cellular damage by activating innate and adaptive immune responses, leading to T-cell exhaustion and immune inhibitory receptor expression (e.g., PD-1, CTLA-4, TIM-3).
Rehermann et al. [26]	Innate immune system pathways	Components like dendritic cells, Kupffer cells, and NK cells perpetuate inflammation and hepatocellular injury in chronic HBV infections.
Revill et al. [27]	Host immune response	Host immune response, rather than the virus itself, is responsible for hepatocellular damage, necessitating continuous lifelong treatment.
Lim et al. [28]	Immunosuppressive tumor microenvironment	HBV–HCC creates an immunosuppressive environment within the tumor to mitigate host damage during prolonged viral infection.
He et al. [29]	Cytokine secretion by NK cells	NK cells secrete cytokines like IL-10, TGFβ, IL-4, and IL-13, facilitating fibrogenesis and tumorigenesis.
Tian et al. [30] Chen et al. [31]	NK cell cytotoxicity and survival	HBV impairs NK cell cytotoxicity and survival, promoting chronic hepatocellular injury and epithelial-to-mesenchymal transition.
Borgia et al. [32]	Factors affecting NK cell dysfunction	Factors like severe hypoxia, tumor cell metabolites, inhibitory receptors, and molecules like TGFβ, PGE2, IDO1 contribute to NK cell dysfunction in HBV–HCC.
Wang. et al. [33]	Role of Kupffer cells and monocytes	Kupffer cells and inflammatory monocytes promote the transition from chronic hepatitis to liver cancer by releasing proinflammatory cytokines such as TNFα, IL-6, and MCP-1, and inducing CD8+ T-cell exhaustion in collaboration with myeloid-derived suppressor cells (MDSCs).
Bertoletti et al. [34] Hao, X et al. [35]	CD8+ T-cell antiviral activity	HBV-specific CD8+ T cells produce proinflammatory cytokines and cytotoxic molecules to eliminate infected cells but contribute to sustained hepatic inflammation and HCC development in chronic HBV infection.
Zong et al. [36] Fisicaro et al. [37]	CD8+ T-cell dysfunction	In chronic HBV infection, CD8+ T cells become dysfunctional, expressing exhaustion markers like PD-1, CTLA-4, LAG3, TIM-3, and TIGIT, contributing to defective anti-tumoral surveillance and chronic hepatic inflammation.
Heim et al. [38] Liu. et al. [39]	Cytokine production by exhausted CD8+ T cells	Exhausted CD8+ T cells maintain chronic hepatic inflammation by producing inflammatory cytokines (IFN-α, TNFα, IL-17A, IL-22) in response to continuous HBV antigen stimulation.
Yan. et al. [40] Fu et al. [41]	CD4+ T-cell response	Chronic HBV infection leads to a shift from Th1 to Th17 CD4+ T cells and a reduction in effector cytotoxic CD4+ T cells, contributing to ongoing hepatocellular injury and diminished anti-tumoral surveillance.
Li et al. [42]	Expansion of Treg cells	Treg cells expand during chronic HBV infection, stimulated by cytokines like TGFβ, IL-10, and CCL22, creating an immunosuppressive microenvironment for HCC initiation and progression.
Schollmeier et al. [44]	HBx protein and hepatocarcinogenesis	HBx protein disrupts hepatocellular cycle regulation, affecting cell proliferation, death, transcription, and DNA repair pathways, and interacts with cellular signaling pathways like CREB, Ras/Raf, MAPK, and JAK-STAT, contributing to genetic instability and neoplastic transformation.
Datfar et al. [45] Levrero et al. [46] Ling et al. [47]	HBx protein and cellular signaling pathways	HBx protein modulates CREB-dependent transcription and engages cellular signaling pathways such as Ras/Raf, MAPK, and JAK-STAT.
Popa et al. [48]	HBx protein and apoptotic effects	HBx protein affects proteasomes, mitochondrial proteins, p53, and DDB1, leading to apoptotic effects and contributing to HCC progression and metastasis.
Yang et al. [49]	HBx protein and HCC progression	Studies demonstrate HBx protein’s involvement in HCC progression and metastasis.
Liu et al. [51]	HBc protein and Src/PI3K/Akt pathway	HBc protein promotes hepatocarcinogenesis by enhancing apoptosis resistance and stimulating hepatoma cell proliferation via the Src/PI3K/Akt pathway.
Capasso et al. [52]	HBc and HBx protein synergy	HBc protein synergizes with HBx protein to exacerbate liver cancer progression by suppressing p53 promoter activity.

Abbreviations: cccDNA: covalently closed circular DNA; CCL22: chemokine (C-C motif) ligand 22; CCNA2: cyclin A2; CCNE1: cyclin E1; CREB: cAMP response element-binding protein; CTNNB1: catenin β1; CTLA-4: cytotoxic T-lymphocyte-associated protein 4; DDB1: DNA damage-binding protein 1; HBx protein: hepatitis B virus X protein; HCC: hepatocellular carcinoma; IDO1: indoleamine 2,3-dioxygenase 1; IFN-α: interferon alpha; IL-4: interleukin 4; IL-6: interleukin 6; IL-10: interleukin 10; IL-13: interleukin 13; IL-17: interleukin 17; IL-22: interleukin 22; JAK-STAT: Janus kinase–signal transducer and activator of transcription; KMT2B: lysine methyltransferase 2B; LAG3: lymphocyte activation gene 3; MCP-1: monocyte chemoattractant protein 1; MAPK: mitogen-activated protein kinase; MLL4: myeloid/lymphoid or mixed-lineage leukemia protein 4; NK: natural killer; NTCP: sodium taurocholate cotransporting polypeptide; PD-1: programmed cell death protein 1; PGE2: prostaglandin E2; Raf: rapidly accelerated fibrosarcoma; PI3K: phosphoinositide 3-kinase; Akt: protein kinase B; Ras: rat sarcoma; TERT: telomerase reverse transcriptase; TGFβ: transforming growth factor β; TIGIT: T-cell immunoreceptor with Ig and ITIM domains; TIM-3: T-cell immunoglobulin and mucin domain-containing protein 3; TNFα: tumor necrosis factor α; TP53: tumor protein P53.

### 2.2. Hepatitis C Infection

Chronic HCV infection is the most prevalent underlying liver ailment among HCC patients globally, notably in regions like North America, Europe, and Japan [53]. The genesis of HCC in association with HCV follows a prolonged trajectory, often spanning decades of chronic infection, subsequent inflammation, and fibrosis [54,55]. Current evidence from multiple studies underscores HCV as an independent risk factor for HCC [56] (Table 2).

Following the initiation of HCV replication within hepatocytes, viral proteins and replication intermediates are detected by pattern recognition receptors (PRRs) as pathogen-associated molecular patterns (PAMPs). This recognition prompts the production of cytokines and chemokines like interferon (IFN)-λ1, interleukin (IL)-1β, and macrophage inflammatory protein (MIP)-1, initiating both innate and adaptive immune responses [57].

In contrast to HBV, HCV exerts its effects on the host genome indirectly and establishes chronic infection in 70–80% of cases. Viral replication does not entail integration into the genome; rather, it occurs surreptitiously through the synthesis of nonstructural proteins (NS; e.g., NS2, NS3, NS4A, NS4B, NS5A, and NS5B) that disrupt the mitogen-activated protein kinase (MAPK) cellular signaling pathway. This disruption leads to the inhibition of apoptosis, cell shrinkage, and fragmentation of the cell nucleus in infected liver cells [58,59]. Therefore, mechanisms such as compromised antigen presentation, upregulated T-cell depletion markers leading to T-cell degradation and increased regulatory T-cell activity may contribute to immune system inefficiency and compromise antiviral function in the context of cancer [60].

Finally, in contrast to HBV, HCV demonstrates its oncogenic potential through its structural proteins, which have been associated with HCC development [61]. Specifically, the core protein possesses the capacity to manipulate intracellular pathways, including the activation of the NF-κB pathway [62], and to enhance the expression of various cellular proteins, such as IL-6 and STAT3. The dysregulation of these proteins induces transformative alterations in hepatocytes [61], and the disrupted immune responses within the inflamed liver microenvironment may undermine the efficacy of anti-tumor immunity, facilitating tumor immune evasion [25].

Specifically, the simultaneous replication of the virus along with inflammatory cytokines and excessive accumulation of extracellular matrix components (e.g., laminin, collagen, fibronectin) contribute to the transition from HCV-associated hepatitis to liver cirrhosis by promoting the differentiation of HSCs into myofibroblasts [63]. Notably, within the viral milieu, these viral proteins enhance both innate and adaptive immune responses by inducing overexpression of platelet-derived growth factor (PDGF), epidermal growth factor (EGF), vascular endothelial growth factor (VEGF), C-X-C motif chemokine ligand 5 (CXCL5), and C-X-C motif chemokine ligand 9 (CXCL9) [64]. In response to the virus, the immune system mounts an intense cytokine response and generates ROS, thereby promoting genomic instability and carcinogenesis. This likely accounts for the persistent risk of hepatocarcinogenesis in patients with cirrhosis and HCV infection, even after the infection has been controlled, a timeframe that warrants further investigation [65].

Moreover, in chronic HCV infection, immune cell dysfunction is evident, as HCV can impede IFN synthesis in effector T cells and induce the expression of exhaustion markers (e.g., TIM-3) on the T-cell surface, thus promoting the onset of HCC in an immunosuppressive environment [66]. Interestingly, chronic HCV infection can also attenuate innate antiviral and anticancer responses by inducing TGFβ-mediated M2-macrophage polarization, IL-100-mediated T reg expansion, and depletion and dysfunction of NK cells [45]. Moreover, chronic HCV infection can promote epithelial–mesenchymal transition, either through the inflammatory hyperactivation of the IL-6–STAT3–lncTCF7 axis, the IL-6–STAT3–Snail–Smad3–TGFβ1 axis, or directly through NS5A-mediated Wntβ catenin pathway dysregulation [67].

**Table 2 ijms-25-07191-t002:** Molecular pathways associated with HCV infection and its progression to liver cancer.

Study	Molecular Pathway	Description
Rosen et al. [57]	Pattern recognition receptors (PRRs) and cytokine production	HCV replication detected by PRRs as PAMPs triggers cytokine and chemokine production (e.g., IFN-λ1, IL-1β, MIP-1), initiating innate and adaptive immune responses.
Ghouri et al. [59]	Nonstructural proteins and MAPK pathway	HCV nonstructural proteins (e.g., NS2, NS3, NS4A, NS4B, NS5A, NS5B) disrupt the MAPK cellular signaling pathway, inhibiting apoptosis and causing cellular changes in infected liver cells.
Zeng et al. [60]	Compromised immune function	HCV compromises antigen presentation, increases T-cell depletion markers, and enhances regulatory T-cell activity, leading to immune inefficiency and compromised antiviral function.
Selimovic et al. [61]	Structural proteins and oncogenesis	HCV structural proteins, especially the core protein, manipulate intracellular pathways (e.g., NF-κB pathway) and enhance expression of proteins like IL-6 and STAT3, contributing to transformative changes in hepatocytes and disrupted immune responses.
Li et al. [62]	NF-κB pathway activation	HCV core protein activates the NF-κB pathway.
Wangensteen et al. [25]	Immune evasion in inflamed liver microenvironment	Disrupted immune responses within the inflamed liver microenvironment undermine anti-tumor immunity and facilitate tumor immune evasion.
Irshad et al. [63]	Hepatic stellate cells (HSCs) activation	HCV and inflammatory cytokines promote the differentiation of HSCs into myofibroblasts, contributing to liver cirrhosis.
Sahin et al. [64]	Cytokine and growth factor overexpression	Viral proteins induce overexpression of PDGF, EGF, VEGF, CXCL5, and CXCL9, enhancing both innate and adaptive immune responses.
Dolina et al. [66]	T-cell dysfunction and exhaustion	Chronic HCV impedes IFN synthesis in T cells and induces exhaustion markers (e.g., TIM-3), promoting HCC in an immunosuppressive environment.
Datfar et al. [45]	M2-macrophage polarization and NK cell dysfunction	Chronic HCV induces TGFβ-mediated M2-macrophage polarization, IL-10-mediated Treg expansion, and NK cell depletion and dysfunction, attenuating antiviral and anticancer responses.
Gurzu et al. [67]	Epithelial–mesenchymal transition (EMT)	HCV promotes EMT through pathways like IL-6/STAT3/lncTCF7, IL-6/STAT3/Snail–Smad3/TGFβ1, and NS5A-mediated Wntβ catenin pathway dysregulation.
Heredia-Torres et al. [68]	Oxidative stress, insulin resistance, and steatosis	HCV-induced oxidative stress, insulin resistance, and steatosis contribute to HCC development by disrupting lipid metabolism and insulin signaling pathways.
Smirnova et al. [69]	NS5A and core protein effects on ROS and mitochondrial function	NS5A protein elevates ROS by increasing Ca2+ influx into mitochondria, activating NF-κB and STAT3, while core protein disrupts mitochondrial respiratory chain complex I, increasing ROS levels.
Zhao et al. [70]	TGFβ and hepatic stellate cell activation	TGFβ recognition triggers hepatic stellate cell activation and ECM production, leading to liver fibrosis, while EMT facilitates hepatocyte transition to myofibroblasts.

Abbreviations: CXCL5: C-X-C motif chemokine ligand 5; CXCL9: C-X-C motif chemokine ligand 9; ECM: extracellular matrix; EGF: epidermal growth factor; IFN-λ1: interferon λ1; IL-1β: interleukin 1β; IL-6: interleukin 6; IL10: interleukin 10; lncTCF7: long non-coding RNA TCF7; MAPK: mitogen-activated protein kinase; MIP-1: macrophage inflammatory protein 1; NF-kB: nuclear factor kappa B pathway; NK: natural killer; TGFβ1: transforming growth factor β1; NS: nonstructural; PAMPs: pathogen-associated molecular patterns; PDGF: platelet-derived growth factor; ROS: reactive oxygen species; STAT3: signal transducer and activator of transcription 3; TGFβ: transforming growth factor β; TIM-3: T-cell immunoglobulin and mucin domain-containing protein 3; Treg: regulatory T cell; VEGF: vascular endothelial growth factor.

Numerous mechanisms contribute to the progression of hepatocarcinogenesis associated with HCV infection, including oxidative stress, insulin resistance, steatosis, epigenetic modifications, and genetic instability [68]. In particular, oxidative stress and the generation of ROS contribute to the upregulation of proinflammatory gene expression, exacerbating the process of carcinogenesis. Notably, the NS5A protein of HCV elevates ROS levels by enhancing the influx of calcium into mitochondria, thereby activating NF-κB and STAT3, which further aggravates oxidative stress [69]. Additionally, the core protein disrupts complex I formation in the mitochondrial respiratory chain by interacting with prohibitin, leading to increased ROS levels [68,69].

Induced by HCV, hepatic steatosis amplifies the risk of HCC development by disrupting lipid metabolism and insulin signaling pathways. Interactions between HCV proteins and these pathways contribute to insulin resistance and steatosis [68]. Activation of HSCs, triggered by TGFβ recognition, promotes liver fibrosis by inducing extracellular matrix (ECM) production, similar to hepatocytes following epithelial–mesenchymal transition (EMT). EMT, induced by factors such as transforming growth factor β1 (TGFβ1) produced during HCV infection, facilitates the transition of hepatocytes to myofibroblasts, leading to ECM production and liver fibrosis [70]. HCV proteins like NS5A and core additionally stimulate EMT through diverse signaling pathways [68].

Inflammatory processes and EMT intersect with intermediary pathways, further shaping a modified microenvironment conducive to carcinogenesis [68].

### 2.3. Hepatitis Delta Infection

The precise mechanism underlying the development of HCC in HBV/HDV co-infection, whether it arises from the combined influence of both viruses, the presence of underlying cirrhosis, or a direct oncogenic effect of HDV [71], remains unclear.

Fattovich et al. [72] found a threefold increase in the risk of HCC among patients with HBV/HDV-related cirrhosis compared to those with HBV-related cirrhosis. Notably, among individuals positive for the HBs antigen with liver disease progression, HDV was implicated in 20% of HCC cases [73]. A recent meta-analysis further illustrated a higher prevalence of HCC associated with HBV–HDV co-infection compared to HBV mono-infection [74].

The mechanism underlying HDV-induced carcinogenesis remains unclear. Initial investigations suggest that HDV may influence the TGFβ pathway associated with fibrosis, thereby accelerating the progression of co-infected cells and serving as a regulatory factor in fibrosis and hepatocarcinogenesis [75].

Moreover, large delta antigen (L-HDAg) may contribute to hepatocarcinogenesis by activating the TGFβ pathway, particularly through the Smad3 protein [73]. Additionally, L-HDAg has been demonstrated to induce NF-κB activation and ROS generation, thereby enhancing the expression of TNFα [45,71,76,77].

## 3. Metabolic Etiological Factors Associated with Inflammation and Hepatocellular Carcinoma

### 3.1. Metabolic Dysfunction-Associated Steatotic Liver Disease (MASLD)

The prevalence of metabolic dysfunction-associated steatotic liver disease (MASLD), formerly termed non-alcoholic fatty liver disease (NAFLD), is increasing, contributing to a rise in advanced liver conditions, including liver fibrosis, cirrhosis, and hepatocellular carcinoma (HCC) [78,79]. Recent data indicate that MASLD, especially in its progression to advanced steatohepatitis (MASH), is becoming the primary driver of the anticipated increase in HCC cases in coming years. Notably, the contribution of MASH to HCC incidence has risen significantly, underscoring the fact that chronic hepatic inflammation serves as a potential catalyst for hepatocarcinogenesis, irrespective of the presence of cirrhosis [80,81]. This multifaceted process involves various risk factors, including genomic instability, obesity, and diabetes mellitus, among others. Notably, the patatin-like phospholipase domain-containing 3 (*PNPLA3*) gene variant (rs738409 c.444 C>G, p.I148M) on chromosome 22 has been linked to impaired triglyceride mobilization and increased hepatic lipid accumulation, thereby elevating the risk of HCC independently of other factors [82,83]. In fact, excessive triglyceride accumulation (e.g., steatosis) leads to hepatocyte lipotoxicity, lipid peroxidation, and stress in the mitochondria and endoplasmic reticulum. This process generates ROS and inflammatory cytokines (e.g., TNFα and IL-6), which in turn promote tumorigenesis by causing DNA mutations and oncogenic transformation [84]. Similarly, the transmembrane 6 superfamily member 2 (*TM6SF2*) gene polymorphism (rs58542926, c.449 C>T, p.E167K) on chromosome 19 is associated with steatosis severity and fibrosis progression in MASH, although its direct role in hepatocarcinogenesis remains debated [85]. Recent findings also suggest that a loss-of-function variant in the hydroxysteroid 17 beta dehydrogenase 13 (*HSD17B13*) gene may reduce the risk of chronic liver disease and the progression from steatosis to steatohepatitis, potentially affecting HCC development [86]. Furthermore, mutations in the homeostatic iron regulator (*HFE*) gene have been linked to a complicated course of MASH, predisposing individuals to HCC development [78].

The context of MASH represents an inflammatory milieu, but while virus-specific CD8 T cells contribute to liver damage in chronic viral hepatitis, no specific antigen or pathogen-driven T-cell response has been shown to cause liver damage in MASH.

Recently discovered CD8 + CXCR6 + PD-1+ T cells capable of auto-aggressive hepatocyte killing were found to initiate NASH and facilitate the transition to liver cancer [87]. In fact, interleukin 15 (IL15) produced in the hepatic microenvironment downregulates forkhead box O1 (FOXO1) in CD8 T cells, enabling them to acquire a resident character by upregulating CXCR6, leading to steatohepatitis (MASH), fibrosis, and facilitating HCC development [88].

### 3.2. Obesity

A similar discussion regarding MASLD is found in the context of obesity, which itself causes MASLD. Notably, obesity-induced alterations in metabolic and immune signaling pathways significantly impact the function of immune cells, particularly CD8+ and CD4+ T cells, in the development of HCC [89,90,91]. Elevated levels of leptin, a hormone associated with obesity, promote glycolysis in CD8+ T cells via the PI3K/Akt/mTOR pathway, exacerbating chronic inflammation linked to obesity [91]. Upon activation, CD8+ T cells undergo metabolic reprogramming, shifting from oxidative phosphorylation to aerobic glycolysis. Despite these adaptive changes, CD8+ T cells become more susceptible to glucose deprivation in the tumor immune microenvironment (TIME), leading to diminished production of crucial effector molecules such as IFN-γ, granzyme B, and perforin, ultimately impairing their anticancer functions [92,93].

Elevated leptin levels and the accumulation of unsaturated fatty acids typical of obesity lead to alterations in CD4+ T cells, resulting in their dysfunction and eventual apoptosis [94]. The activation of the peroxisome proliferator-activated receptor alpha (PPAR-α) pathway due to increased fatty acid availability further exacerbates these effects, impairing CD4+ T-cell function and promoting liver carcinogenesis [95]. In particular, linoleic acid causes mitochondrial dysfunction, leading to the selective loss of CD4+ T cells in the liver and fostering a pro-carcinogenic environment [96].

Given the increasing global obesity rates, further studies are needed to fully understand how obesity and its associated inflammation contribute to the initiation of hepatocarcinoma development.

## 4. Alcohol Etiological Factors Associated with Inflammation and Hepatocellular Carcinoma

Alcohol is classified as a group 1 carcinogen by the International Agency for Research on Cancer, as it has been reported to be responsible for approximately 15–30% of HCC, even at doses as low as 10 g/1 unit/day, with a relative risk of 2.07 for heavy and occasional drinkers compared to non-drinkers [97,98].

Alcoholic liver disease includes various types of histological liver abnormalities, ranging from steatosis to steatohepatitis to cirrhosis, and is an important contributor to hepatocarcinogenesis synergistically with MASLD and viral infections [99]. Additionally, inflammation caused by alcohol has a strong association with the severity of disease and negative outcomes in patients with HCC [100]. Chronic consumption of alcohol results in the excessive buildup of fat (primarily triglycerides, phospholipids, and cholesterol esters) within liver cells, leading to changes in the structure and functional ability of the liver. This process triggers the development of steatosis and steatohepatitis. Moreover, the prolonged intake of excessive amounts of ethanol (such as 60–80 g/day of alcohol in men and 20 g/day in women) has been linked to the progression of cirrhosis within a span of 10 years. This in itself poses a significant risk factor for the development of hepatocarcinogenesis, regardless of the combined impact of hepatitis viruses and metabolic disorders [97,101]. Nevertheless, it may also promote the initiation of HCC through various alternative molecular pathways [102]. The formation of acetaldehyde, which in turns produces carcinogenic DAN adducts, the overexpression of cytochrome P450 family 2 subfamily E member 1 (CYP2E1)-induced reactive oxygen and nitric oxygen species (ROS and NOS) as well as impairment in the antioxidant defenses and DNA repair mechanisms and altered gene transcription are the most involved pathways [100]. Firstly, acetaldehyde formed by ethanol metabolism through alcohol dehydrogenase (ADH) creates various protein and DNA adducts that promote DNA repair failure, lipid peroxidation, and mitochondrial damage, thus favoring carcinogenesis [97]. Secondly, the overaccumulation of ROS produced by CYP2E1 as an alcohol effect can either alter gene expression or foster an exaggerated liver inflammatory response [103]. Moreover, an increased neutrophil count in the liver mediated by the IL-8–IL-13 axis [104], the overproduction of IL-17 by Kupffer cells and by CD4+ Th cells [105], the mitochondrial oxidative damage, the endoplasmic reticulum stress, and the unfolded protein response driven by ethanol overload can synergistically maintain liver inflammation and foster carcinogenesis [99,106]. In fact, the persistent liver inflammation leads to the activation of programmed cell death pathways, thus favoriting hepatocyte cell death and compensatory proliferation of surviving mutated clones [107].

Thirdly, alcohol may interact with the immune system and affect tumor immune surveillance indirectly via impairing gut permeability and facilitating the translocation of bacteria-derived lipopolysaccharides (LPSs) and other bacterial products from the gut to the liver, where they could elicit a dangerous proinflammatory and pro-carcinogenic response mediated by TLR4–IL6–TNFα activation of the STAT3–NF-κB axis in innate immune cells [108].

Moreover, pathological changes in gut microbiota composition with decreased levels of anti-inflammatory bacteria such as *Faecalibacterium prausnitzi* and *Bidovacterium* spp. and increased levels of proinflammatory strains such as Proteobacteria spp. can foster inflammation in alcoholic liver diseases [109]. In this regard, fecal microbiota transplantation and fecal microbiota manipulation via use of prebiotics might represent a valuable therapy to prevent alcohol-induced hepatocarcinogenesis [110].

In conclusion, although abstinence from alcohol is necessary to prevent HCC, its effects may persist for decades, as a washout period of 23 years is required to reduce the incidence of HCC to the same level as in abstainers [111].

## 5. The Contribution of the Gut–Liver Axis to Hepatocarcinogenesis through Inflammation

The gut–liver axis plays a pivotal role in maintaining systemic inflammation in liver diseases. Conditions like cirrhosis and MASLD often alter the gut barrier, increasing its permeability. This “leaky gut” condition allows bacteria and their products, such as lipopolysaccharides (LPSs), to enter the bloodstream. These endotoxins, reaching the liver via the portal vein, trigger immune responses that exacerbate liver inflammation and damage [112] (Table 3).

The activation of the LPS/TLR4 signaling pathway in Kupffer cells leads to TNFα- and IL-6/STAT3-dependent hepatocyte compensatory proliferation. Additionally, TLR4 signaling in hepatic stellate cells upregulates NF-κB/p65-dependent epiregulin, both of which inhibit physiological hepatocyte apoptosis [113]. This signaling can also activate the inflammatory MAPK4 (MKK4)/c-Jun N-terminal kinase (JNK) pathway, stimulating the production of MMP2, MMP9, IL-6, IL-23, IL-17A, and TNFα, thereby fostering EMT and HCC invasiveness [114] (Table 3).

Liver diseases often cause dysbiosis, an imbalance in gut microbiota that increases harmful metabolites and decreases beneficial ones, promoting gut permeability and systemic inflammation. Beneficial metabolites like short-chain fatty acids (SCFAs), which have anti-inflammatory effects, are reduced in dysbiosis, diminishing their protective effects. Altered bile acid metabolism in liver diseases leads to toxic bile acid accumulation, damaging the gut lining and disrupting the microbiota [115]. These bile acids interact with gut receptors, such as the farnesoid X receptor (FXR), which regulates inflammation and gut barrier integrity. Dysregulation of this pathway exacerbates gut and systemic inflammation [116].

The gut contributes to chronic liver inflammation through bile acid metabolism alterations, causing hepatocyte apoptosis, mitochondrial and endoplasmic reticulum stress, ROS generation, and the overproduction of proinflammatory cytokines (e.g., TNFα, IL-1β, and IL-6) via p38/MAPK-dependent activation of p53 and NF-κB [113]. Downregulation of FXR by gut-mediated toxic bile acid overload and proinflammatory cytokines in chronic liver diseases is a key factor in hepatocarcinogenesis, as FXR normally acts as a potent oncosuppressor by suppressing local inflammation [117,118].

Deoxycholic acid overproduction in gut dysbiosis promotes HCC progression by activating HSCs to secrete proinflammatory cytokines (e.g., IL-6, PGE2, CXC9), which create an immunosuppressive milieu by suppressing NK and T-cell responses, Short-chain fatty acids also play a role in favoring HBx-mediated carcinogenesis [119,120].

An overall shift in gut microbiota composition from beneficial bacteria to proinflammatory LPS- and bile acid-producing genera, such as *Bacteroides*, *Lachnospiraceae incertae sedis*, and *Clostridium XIVa*, is observed during cirrhosis and early HCC [121,122,123].

**Table 3 ijms-25-07191-t003:** Molecular pathways associated with gut–liver axis and its progression to liver cancer.

Study	Molecular Pathway	Description
Luo et al. [113]	LPS/TLR4 signaling pathway	Activation in Kupffer cells leads to TNFα- and IL-6/STAT3-dependent hepatocyte compensatory proliferation. In hepatic stellate cells, TLR4 signaling upregulates NF-κB/p65-dependent epiregulin, inhibiting apoptosis.
Kang et al. [114]	Inflammatory MAPK4 (MKK4)/c-Jun N-terminal kinase (JNK) pathway	Activation stimulates production of MMP2, MMP9, IL-6, IL-23, IL-17A, and TNFα, fostering EMT and HCC invasiveness.
Shi et al. [116]	Bile acid metabolism and FXR pathway	Altered bile acid metabolism leads to toxic bile acid accumulation, damaging the gut lining and disrupting the microbiota. Bile acids interact with FXR, which regulates inflammation and gut barrier integrity.
Luo et al. [113]	p38/MAPK-dependent activation of p53 and NF-κB	Bile acid metabolism alterations cause hepatocyte apoptosis, mitochondrial and ER stress, ROS generation, and the overproduction of proinflammatory cytokines (e.g., TNFα, IL-1β, IL-6).
Huang et al. [117] Wei et al. [118]	FXR pathway and oncosuppression	Downregulation of FXR by toxic bile acid overload and proinflammatory cytokines in chronic liver diseases is key in hepatocarcinogenesis. FXR normally suppresses local inflammation.
Chen et al. [119] Jia et al. [120]	Deoxycholic acid and SCFA in HCC progression	Deoxycholic acid overproduction promotes HCC by activating HSCs to secrete proinflammatory cytokines, creating an immunosuppressive milieu. SCFAs favor HBx-mediated carcinogenesis.
Ren et al. [121] Huang et al. [122] Zheng et al. [123]	Gut microbiota shift in liver diseases	Shift from beneficial bacteria to proinflammatory LPS- and bile acid-producing genera (e.g., Bacteroides, Lachnospiraceae incertae sedis, Clostridium XIVa) is observed during cirrhosis and early HCC.

Abbreviations: EMT: epithelial-mesenchymal transition; ER: endoplasmic reticulum; FXR: farnesoid X receptor; HCC: hepatocellular carcinoma; HSCs; hepatic stellate cells; IL-6/STAT3: interleukin 6/signal transducer and activator of transcription 3; IL-6: interleukin 6; IL-17A: interleukin 17A; IL-23: interleukin 23; JNK: c-Jun N-terminal kinase; LPS/TLR4: lipopolysaccharide/Toll-like receptor 4; MAPK4: mitogen-activated protein kinase 4; MMP2: matrix metalloproteinase 2; MMP9: matrix metalloproteinase 9; NF-kB: nuclear factor kappa B; ROS: reactive oxygen species; TNFα: tumor necrosis factor α.

In a recent study led by Jinato T. et al. [124], researchers noted distinct differences in the gut microbiota of healthy individuals compared to those with HCC. The findings indicated that healthy controls had lower levels of *Romboutsia*, UCG-002, *Eubacterium hallii*, Erysipelotrichaceae UCG-003, Lachnospiraceae NK4A-136, Lachnospiraceae ND-3007, and *Bilophila*. In contrast, HCC patients showed higher levels of Bacteroides, Streptococcus, *Veillonella*, *Ruminococcus gnavus*, and *Erysipelatoclostridium*.

These findings are consistent with those reported by Ponziani et al. [125], where patients with hepatocellular carcinoma (HCC) exhibited higher levels of Bacteroidetes, *Bacteroidaceae*, *Streptococcaceae*, *Enterococcaceae*, *Gemellaceae*, *Phascolarctobacterium*, *Enterococcus*, *Streptococcus*, *Gemella*, and *Bilophila*. Ren et al. [121] also demonstrated increased levels of Klebsiella and Haemophilus in HCC patients compared to healthy controls, who showed elevated levels of *Ruminococcus*, *Oscillibacter*, *Faecalibacterium*, *Clostridium IV*, and *Coprococcus*. Conversely, Behary et al. [126] observed a reduction in *Oscillospiraceae* and *Erysipelotrichaceae* among HCC patients.

Therefore, these findings suggest the need for larger samples in future clinical studies aimed at analyzing the variations in individual components of the intestinal microbiota. This could aid in identifying tumor markers, as proposed by Gok Yavuz B et al. [127], and potentially considering them as therapeutic targets.

A study on a rat HCC model demonstrated that restoring gut homeostasis with probiotics reduced inflammation, fibrosis and HCC development [128]. The antiviral activity of probiotics may reduce HCC risk by preventing chronic liver inflammation and modulating oncogene and oncosuppressor expression [129]. Furthermore, microbial signatures can influence immunotherapy response, though the exact mechanisms require further elucidation [130]. Whether alterations in the gut microbiota in HCC can be permanently restored by fecal microbiota transplantation remains under investigation [126,131].

## 6. Inflammation and Treatment of Hepatocellular Carcinoma: A Double-Edged Weapon

Inflammation acts as a “double-edged sword” in the progression of HCC. While chronic inflammation is a major factor promoting tumor progression and immune evasion, it can also lead to the inflammation-related death of tumor cells, either directly through ferroptosis or indirectly by activating anticancer immune responses, thereby suppressing HCC progression [132].

During the transformation from inflammation to cancer, there is a suppression of effector immune cells and a recruitment of immunosuppressive cells. This results in an imbalance between the tumoricidal effector response and the tolerogenic immune response. The evolution of immune evasion in HCC involves the expansion of immunosuppressive cells, failure in tumor antigen presentation and processing, an imbalance between proinflammatory and anti-inflammatory cytokines, alterations in immune checkpoint pathways, and interactions between immune inhibitory receptors and their ligands [133].

Moreover, inflammation can negatively affect the effectiveness of anticancer therapy, as systemic inflammation represents an independent risk factor for HCC onset, progression, and resistance to therapies [134,135]. Inflammation can hamper CD8+ T-cell anti-tumor response in two primary ways. First, inflammatory cytokines (e.g., IL-6, IL-17, IL-25, TNFα, and EGF) can directly induce programmed cell death ligand-1 (PD-L1) expression by activating signaling pathways such as interferon, PI3K–AKT, MEK–ERK, JAK–STAT, c-MYC, and NF-kB, leading to T-cell exhaustion [136]. Second, inflammatory molecules like PGE2, nitric oxide, TGFβ, and IL-10 can recruit immunosuppressive cells, which in turn dampen CD8+ T-cell killing ability [137].

Systemic treatment has historically played a relatively modest role in managing HCC due to the lack of effective agents, limited survival benefits, and the lower efficacy of tyrosine kinase inhibitors (TKIs) compared to immune checkpoint inhibitors in viral-related HCC [138]. However, clinical data from subgroup analyses of several phase III trials suggest that ICI-based therapy is more effective than the control arm (TKI or placebo) in patients with HCC associated with underlying viral liver disease compared to non-viral etiologies [139,140]. This increased efficacy is likely due to the loss of anti-tumor CD4+ T cells and the accumulation of exhausted, unconventionally activated CD8+ PD-1+ T cells in MASH, which impair tumor immune surveillance and reduce immunotherapy efficacy [141]. Additionally, the accumulation of proinflammatory macrophages in the liver in an MASLD environment and alterations in the gut microbiota may also act as immunosuppressive factors [126].

Currently, there are several therapeutic regimens for treating hepatocellular carcinoma (HCC). The combination of atezolizumab and bevacizumab was the first immune checkpoint inhibitor-based therapy for unresectable advanced HCC, established as the new standard of care in systemic first-line treatment following a successful phase III trial [139]. Another combination, sintilimab plus bevacizumab, is still being evaluated as a systemic first-line therapy primarily for HBV-associated HCC in a Chinese phase II/III trial [142]. The combination of durvalumab and tremelimumab has already been included as a first-line option in the latest HCC treatment recommendations [143]. Since 2020, the combination of nivolumab and ipilimumab has been approved by the FDA for advanced HCC patients who have previously received sorafenib [144]. The rationale for combining immune checkpoint inhibitors with anti-vascular endothelial growth factor (VEGFR) antibodies is based on the involvement of the VEGF pathway signaling in diminishing anti-tumoral immunity through several mechanisms, which include reducing the cytotoxic activity of peripheral T cells, promoting CD8+ T-cell apoptosis, enhancing T-regulatory cell activation, and inducing myeloid-derived suppressor cells and M2 macrophages. These effects collectively lead to immunosuppression by depleting lymphocytes, generating oxidative stress, and activating T-regulatory cells [145].

During immune checkpoint inhibition therapy, the tumor microenvironment is remodeled, transforming tumor-promoting inflammation into tumor-suppressive inflammation. Ferroptosis, a specific form of cellular death dependent on lipid peroxidation, is commonly observed in cancer cells undergoing radiotherapy, chemotherapy, and immunotherapy [146,147]. Lipid peroxides produced during ferroptosis stimulate the processing of tumor antigens by dendritic cells, activating cytotoxic T cells to eliminate tumor cells and consequently enhancing the effectiveness of immunotherapy [147]. Collaterally, a recent study by Ryan et al. identified a genetic signature in HCC, including alcohol dehydrogenase 4 (ADH4), apolipoprotein 5 (APOA5), complement factor H-related 3 (CFHR3), C-X-C motif chemokine ligand 8 (CXCL8), formimidoyltransferase cyclodeaminase (FTCD), glucose-6-phosphate dehydrogenase (G6PD), and paraoxonase-1 (PON1), which could potentially predict a better response to ICIs due to the activation of local acute inflammation and anticancer immunity [132]. Another study by Gao et al. identified an inflammatory response-related gene signature model, involving meprin A subunit alpha (*MEP1A*), C-C motif chemokine ligand 20 (*CCL20*), adenosine A2b receptor (*ADORA2B*), tumor necrosis factor superfamily member 9 (*TNFSF9*), intercellular adhesion molecule 4 (*ICAM4*), and solute carrier family 7 member 2 (*SLC7A2*), to predict prognosis and response to immunotherapy in virus-related HCC [148].

In patients who are not eligible for immune checkpoint inhibitors, sorafenib and lenvatinib remain the first-line therapies for unresectable HCC. A novel approach involves combining sorafenib with loco-regional therapies such as transarterial chemoembolization (TACE). TACE can cause acute hypoxia and upregulation of VEGF, which may lead to revascularization. Inhibiting platelet-derived growth factor receptor (PDGFR) and VEGFR-1, -2, and -3 with sorafenib may help prevent tumor survival and spread and alter the immunosuppressive microenvironment [149].

A potential future therapy for viral-related hepatocellular carcinoma (HCC) could involve the use of engineered autologous “CAR T cells” that express chimeric antigen receptors (CARs) targeting viral tumor antigens [150]. However, further studies are needed to assess their clinical applicability. Given the role of epigenetic modifications in influencing the expression of proto-oncogenes and tumor suppressors, other novel anticancer therapies such as histone deacetylase inhibitors (HDACis) are being evaluated in clinical trials for unresectable HCC, either alone (NCT00321594) or in combination with sorafenib (NCT00943449) [151,152]. Epigenetic drugs can reactivate T cells and promote tumor cell death by inducing the expression of death receptors, stress-induced ligands, and co-stimulatory molecules, making tumor cells more susceptible to immune-mediated lysis. This sensitization enhances the effectiveness of immune checkpoint therapies targeting PD-L1, PD-L2, PD-1, and CTLA-4 [137].

Orthotopic liver transplantation (OLT) is the only effective treatment option for early-stage unresectable HCC. However, the inflammatory tumor microenvironment, sustained by allograft ischemia and reperfusion (IR) injury, stimulates the expression of IL-6, IL-5, TNF, CXCL10, and CCL2. Combined with post-transplant immunosuppression, this environment can drive oncogenesis and HCC recurrence [153].

## 7. Conclusions

Hepatocellular carcinoma (HCC) arises from inflammatory stimuli, with inflammation playing a pivotal role in both its initiation and progression. Cirrhosis epitomizes the apex of chronic liver inflammation, where a cascade of cytokines and immune cell activity exacerbates fibrosis, promotes neoangiogenesis, and compromises immune defenses, fostering an environment conducive to the proliferation of cancer stem cells and the invasiveness of tumor cells. However, non-cirrhotic conditions such as MASLD and viral infections remain significant contributors to carcinogenesis.

Viruses can directly induce genetic instability in host cells or disrupt normal cell cycles through their oncogenic proteins. Indirectly, they heighten oxidative stress, promoting aberrant cellular survival. Following malignant transformation, sustained disruptions in regulatory immune cell functions perpetuate a state of low-grade local and systemic inflammation, impairing secondary surveillance mechanisms crucial for eliminating abnormal cells. This creates a self-perpetuating cycle.

Furthermore, inflammation can undermine the efficacy of anticancer therapies. Consequently, eliminating the underlying inflammatory triggers, whether viral or not, may lead to crucial adjunctive strategies in HCC management. Systemic inflammation emerges as an independent risk factor for HCC onset, progression, and resistance to anticancer treatments. Standard markers of systemic inflammation, such as the neutrophil:lymphocyte ratio (NLR), platelet:lymphocyte ratio (PLR), and prognostic nutritional index (PNI), hold promise for predicting HCC prognosis and tumor response to immune checkpoint inhibitors, yet therapeutic strategies targeting inflammatory cells and signaling pathways remain largely unexplored in this context.

## Figures and Tables

**Figure 1 ijms-25-07191-f001:**
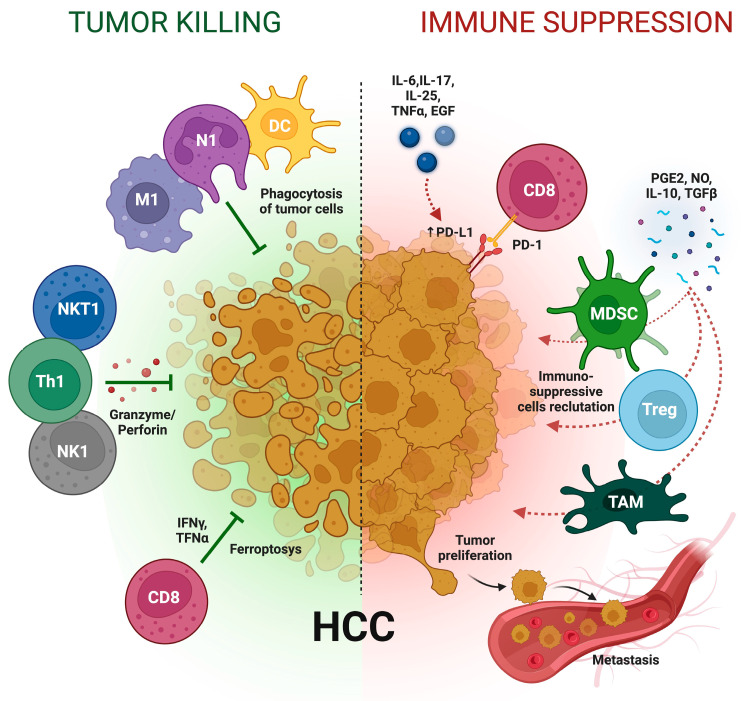
The complex interplay of viral and non-viral inflammatory response in the pathogenesis and treatment of hepatocellular carcinoma. Abbreviations: CD8: CD8 T lymphocytes; DC: dendritic cell; EGF: epidermal growth factor; HCC: hepatocellular carcinoma; IL: interleukin; IFNγ: interferon γ; M1: M1 macrophages; MDSC: myeloid-derived suppressor cells; N1: N1 neutrophil; NK1: type 1 natural killer; NKT1: type 1 natural killer T cell; NO: nitric oxide; PD-1: programmed cell death protein 1; PD-L1: programmed death ligand 1; PGE2: prostaglandin E2; TAM: tumor-associated macrophage; TGFβ: tumor growth factor β; Th1: type 1 T helper; TNFα: tumor necrosis factor α; Treg: regulatory T cell.

## Data Availability

Not applicable.

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
