# Peer review of "Inflammatory Response in the Pathogenesis and Treatment of Hepatocellular Carcinoma: A Double-Edged Weapon"

_ijms, 2024, doi:10.3390/ijms25137191_

Round 1

Reviewer 1 Report

Comments and Suggestions for Authors

see attached

Comments on the Quality of English Language

Minor editing suggested.

Reviewer 2 Report

Comments and Suggestions for Authors

The review article entitled "Inflammatory response in the pathogenesis and treatment of hepatocellular carcinoma: a double-edged weapon" submitted by Galasso et.al. discusses the role of inflammation in the pathogenesis of hepatocellular carcinomas as the inflammatory stimuli give rise to HCC initiation and progression. The review point outs the paramount importance of the liver cirrhosis during the chronic liver inflammation. The authors discuss the role of cytokines and the immune mediator's role in the different stages of HCC namely fibrosis, angiogenesis, compromised immune defences, cancer stemness and invasiveness. Apart from this the authors also emphasises the role of viral infections in the HCC and the mechanisms by which it can induce the cancer.  The authors also discuss the importance of systemic inflammatory markers like neutrophile-lymphocyte ratio, platelet lymphocyte ratio and prognostic nutritional index in understanding the prognosis of HCC and the treatment response. The review is well written with detailed sections within the topic and elaborating each section. However there are few points which needs to be addressed which are as follws.

The role of DAMPs in the initiation of HCC is recognized and how it initiated the by the PRRs is well documented in (PMID: 29872724) and the importance of DAMPs in the fate of tumor is also recognized in (PMID: 29032985). The authors should mention this topic and cite them in the review to give an appropriate picture.     The HCC-associated inflammation can be initiated and propagated by extrinsic pathways through activation of pattern-recognition receptors (PRRs) by pathogen-associated molecule patterns (PAMPs) derived from gut microflora or damage-associated molecule patterns (DAMPs) released from dying liver cells.

Overall, the review is well written and gives a thorough understanding of the area.

Reviewer 3 Report

Comments and Suggestions for Authors

Hepatocellular carcinoma may develop in chronically inflamed hepatic milieu. Chronic hepatitis B and C are currently major risk factors for HCC, causing liver damage through direct and immune-mediated mechanisms. MASLD is also an emerging risk factor, driven by obesity and insulin resistance, and promotes inflammation and carcinogenesis. The gut-liver axis also contributes to HCC through gut barrier disruption and dysbiosis, leading to increased inflammation and liver damage. These conditions promote tumor growth by inducing genetic instability, oxidative stress, and immune suppression. While chronic inflammation upregulate tumor progression, it can also drive inflammation-related tumor cell death and activate anti-cancer immune responses, highlighting the complex role of inflammation in HCC.
The report addresses multifaceted roles of inflammation HCC, tackling both disease progression and treatment and specific mechanisms through viral infections (hepatitis B, C, and D) and metabolic conditions like MASLD. The report explores the gut-liver axis. And the report emphasizes the complex role of inflammation in HCC treatment, particularly the potential of immunotherapy to release inflammation for tumor suppression while emphasizing its potential to hinder treatment efficacy. This report also discusses emerging research areas, such as the identification of genetic signatures that could predict responses to immunotherapy and the development of novel therapies like CAR T cells and epigenetic drugs, enhancing the perspectives on HCC management.

Comments on the Quality of English Language

none.

Reviewer 4 Report

Comments and Suggestions for Authors

The manuscript by Galasso et al aimed to provide a comprehensive overview of the role of inflammatory response in HCC initiation and management. The topic is relevant and encourages further studies to deeply explore the therapeutic strategies targeting inflammatory cells and signaling pathways. Nevertheless, enhancing clarity and addressing certain corrections could improve its overall quality.

In the Abstract and Introduction sections, the aim of the review should be better explained. Moreover, the Conclusions section should be revised with an in-depth discussion on the role of inflammatory response in HCC, avoiding redundancy of sentences reported in previous paragraphs.

In Tables 1-3, the authors indicated the molecular pathways associated with HBV/HCV infections and the gut-liver axis. However, no supporting references are reported. Moreover, the main molecular pathways should be briefly discussed in the main text. Alternatively, it would be beneficial for the authors to consider including representative images that summarize these molecular pathways and their relationship with HCC progression.

Since Alcoholic Liver Disease and related inflammatory processes represent one of the most common risk factors of HCC, the authors should also discuss this issue in a specific subparagraph.

Page 13 (lines 368-371), the authors reported “Authors should discuss the results and how they can be interpreted from the perspective of previous studies and of the working hypotheses. The findings and their implications should be discussed in the broadest context possible. Future research directions may also be highlighted”. This sentence makes no sense and does not provide additional information regarding obesity involvement in HCC. Please better clarify it.

In paragraph 4, the authors described the contribution of the gut-liver axis to hepatocarcinogenesis through inflammation. In this regard, it would be useful to report recent findings on microbiota profiling in HCC patients compared to healthy individuals to provide strong evidence of microbiota contribution to inflammation and HCC initiation.

Minor comments:

Define abbreviations at first mention and use them consistently throughout the text (e.g. page 3 line 99, page 11 lines 244-245, page 12 line 315, etc).

The extended full name of genes, transcription factors, and proteins should be reported when they are cited for the first time along the main text. Moreover, full gene/transcription factor names and abbreviations must be italicized (e.g. page 2, line 77; page 3, lines 104-106, etc).

The authors should check the manuscript for style and grammar errors. For example, the abbreviation for Metabolic-associated fatty liver disease is MAFLD and not MASLD (page 2, lines 57-59). Moreover, the Hepadnaviridae family must be italicized (page 3 line 99).

Round 2

Reviewer 4 Report

Comments and Suggestions for Authors

The authors properly addressed all the comments. The manuscript may be accepted in present form.